# Poloxamer 407 Based Gel Formulations for Transungual Delivery of Hydrophobic Drugs: Selection and Optimization of Potential Additives

**DOI:** 10.3390/polym13193376

**Published:** 2021-09-30

**Authors:** Kamran Hidayat Ullah, Faisal Raza, Syed Mohsin Munawar, Muhammad Sohail, Hajra Zafar, Mazhar Iqbal Zafar, Tofeeq Ur-Rehman

**Affiliations:** 1Department of Pharmacy, Quaid-i-Azam University, Islamabad 45320, Pakistan; kamran_893@yahoo.com (K.H.U.); syedmohsin013@yahoo.com (S.M.M.); 2School of Pharmacy, Shanghai Jiao Tong University, 800 Dongchuan Road, Shanghai 200240, China; faisalraza@sjtu.edu.cn (F.R.); hajrazafar@sjtu.edu.cn (H.Z.); 3School of Pharmacy, Yantai University, Shandong 264005, China; sohailshah11@yahoo.com; 4Department of Environmental Sciences, Quaid-i-Azam University, Islamabad 45320, Pakistan; mazhariqbal.zafar@gmail.com

**Keywords:** hydrogel, poloxamer 407 polymer, poloxamer 407 gel, transungual drug delivery, onychomycosis, ungual penetration enhancer, Terbinafine

## Abstract

The current study aimed to develop poloxamer 407 (P407) gel for transungual delivery of antifungal hydrophobic drugs with sufficient gel strength and drug loading. Gel strength and drug loading of P407 gel was improved by use of functional additives. Hydration enhancement effect was used to select optimum nail penetration enhancer. Face-centered central composite design (FCCCD) was used to observe the effect of the selected penetration enhancer (thioglycolic acid (TGA)) and cosolvent (ethanol) on gelation behavior to develop formulation with enough loading of hydrophobic drug, i.e., terbinafine HCl (TBN), and its permeation across the nail plate without compromising on gel strength. It was observed that increasing concentration of P407 and TGA significantly reduced gelation temperature and enhanced the gel strength of P407 gel and can be used to improve P407 gel strength. Under the scanning electron microscope, the significant effect of TGA as an ungual penetration enhancer was observed on the morphology of the nail plate. Optimized P407 gel prepared with modified cold method showed a gelation temperature of 8.7 ± 0.16 °C, gel strength of 122 ± 7.5 s and drug loading of 1.2% *w*/*w*, which was four times more than the drug loading in the gels prepared with conventional cold method. Rheological behavior was pseudoplastic with 47.75 ± 3.48% of gel erosion after 12 washings and 67.21 ± 2.16% of drug release after 12 h. A cumulative amount of TBN permeated from P407 gel with and without PE after 24 h was 27.30 ± 4.18 and 16.69 ± 2.31 µg/cm^2^, respectively. Thioglycolic acid can be used as a nail penetration enhancer without the chemical modification or addition of extra additives while retaining the gel strength. Water miscible cosolvents with moderate evaporability such as ethanol, can be incorporated to P407 gel by minor modification in method of preparation to load the required dose of hydrophobic drugs. Developed P407 gel formulation with sufficient gel strength and drug loading will be a promising carrier for transungual delivery of hydrophobic antifungal agents.

## 1. Introduction

Onychomycosis accounts for 50% of nail diseases, which affect 10% of the general population with higher prevalence in people over 60 years of age. Oral systemic antifungal therapy is limited by its hepatotoxicity, drug interactions, a long duration of treatment, high cost of medication, increased microbial resistance and relapse of infection. Topical application of drug to nail is preferable because of its localized effects, minimum adverse effects and less drug interactions [1]. The major limitation of topical ungual drug delivery is low nail permeability due to the presence of highly stable disulfide and hydrogen bonds found in nail keratin [2]. Topical delivery of hydrophilic drugs may be facilitated through hydration of the nail plate and addition of nail penetration enhancers. However, the majority of recently approved drugs, including antifungals, are hydrophobic in nature and solubilization of the required dose is another challenge for transungual delivery of hydrophobic drugs. Nail permeation of drug molecules may be enhanced by the addition of reducing and oxidizing agents [3], keratolytic agents [4], keratinolytic enzymes [5] and surfactants [6].

Nail lacquer, a solution of drug and polymer in organic/vaporizing solvent, is one of the approaches where hydrophobic drugs and layer of the polymer deposits on the nail plate after evaporation of the organic solvent. Ciclopirox nail lacquer, approved by the FDA as topical treatment in onychomycosis, requires another organic solvent for removal which can damage barrier properties of the nail surface [7]. Unattended disposal of lacquers and organic solvents, just as nail polishes, may pose threat to the environment [8]. Moreover, rapid evaporation of organic solvent causes the formation of drug crystals, posing a barrier in its diffusion/permeation into the nail [9].

Polymer based platforms have been successfully used in various biomedical applications [10,11]. P407 gels, due to high water content, may hydrate the nail plate to facilitate permeation across the nail plate and can be a good alternate to nail lacquer for topical delivery of hydrophobic drugs such as terbinafine and efinaconazole [12]. P407 gels have an inherent tendency to improve the solubilization of hydrophobic drugs due to amphiphilic nature of polymer. Moreover, additives of different functionalities may be incorporated in gels without compromising their gelation ability [13,14]. Tanriverdi and Ozer reported the highest amount of drug accumulation in nail using poloxamer gel as compared to chitosan and carbopol gel [15]. However, P407 gels will be retained on the nail plate for a shorter period as compared to nail lacquer due to insufficient gel strength, weak stability and fast dissolution of the gels [16]. Drug loading capacity and gel strength of P407 gels may be improved either with the functional additives or by applying a chemical modification/conjugation approach [16,17,18]. Chemical modification and conjugation with crosslinker/polymers require costly and lengthy toxicity testing. The nail penetration enhancers and cosolvents are essential additives of P407 gels poised for transungual delivery of hydrophobic antifungal drugs and their impact on gel properties must be evaluated thoroughly.

In the present study, topical P407 gel formulation was developed with the aim to enhance the loading of hydrophobic drugs and to improve transungual permeation without compromising on thermogelation and washability of these gels. Nail permeation enhancers were screened based on the hydration enhancement factor [19]. TBN was selected as a model drug due to its strong antifungal activity against dermatophytes [20] and low aqueous solubility. Terbinafine is highly soluble in ethanol which is miscible with water and has a low vaporizing tendency as compared to ethyl acetate and acetone [21]. Optimized P407 gel was selected after assessing the effect of the selected PE (Thioglycolic acid), cosolvent (ethanol) and concentration of p407 on gel properties using face-centered central composite design (FCCCD).

## 2. Materials and Methods

### 2.1. Materials

Urea, tartaric acid, lactic acid, thiourea and oxalic acid were purchased through local supplier from BDH laboratories, Dorset, United Kingdom. Thioglycolic acid (TGA), chitosan, poloxamer P407 (Pluronic F127) and glycolic acid were purchased through local supplier from Sigma-Aldrich, Dorset, United Kingdom. Beta Cyclodextrin was purchased from Cydex Pharmaceuticals, Lenexa, Kansas, USA. Hydroxyl propyl cyclodextrin was supplied by Merck, United Kingdom. DMSO was purchased from Duksan Pure Chemicals, Ansan, Korea. Other chemicals used were of analytical grade.

### 2.2. Nail Clippings and Ethical Approval

This study was performed according to the principles of Declaration of Helsinki and ethical approval was obtained from the Bio-Ethical Committee (BEC) of Quaid-i-Azam University, Islamabad Pakistan (Protocol # BEC-FBS-QAU-2017-37). Human nail clippings were collected from healthy volunteers and informed consent was ensured in all cases.

### 2.3. Screening of Penetration Enhancers

PE were screened on the basis of hydration enhancement effect (HEF) [19]. Briefly, nail clippings, grouped on the basis of size and weight, were placed in 1 mL aqueous solution of each PE and control (deionized water) in a separate glass vial. The vials were sealed and stored at 25 ± 2 °C. After 24 h, nail clippings were removed from glass vials and dried with tissue paper to remove any residual solvent from the nail surface. The nail clippings were then weighed and HEF was calculated using following equation:(1)$$HEF=\frac{weight\;gain\;of\;nail\;clipping\;immersed\;in\;PE\;\left(\%\right)}{weight\;gain\;of\;nail\;clipping\;immersed\;in\;deionized\;water\;\left(\%\right)}\;$$

### 2.4. Topography of Nail

Effect of hydration and selected penetration enhancer on nail morphology was observed by scanning electron microscopy. Samples were sputtered with gold in sputter coater for 90 s at 30 mA. Images were taken by scanning electron microscope (SEM; Joel JSM-5910, Tokyo, Japan) available at the Central Resources Library (CRL), University of Peshawar, Pakistan. Morphology of untreated and treated nail clippings (3 mm central flat portion) was recorded. Treated nail clippings were either immersed in 5% *w*/*v* solution of thioglycolic acid or in deionized water for 72 h.

### 2.5. Preparation of P407 Gel

Previously reported “cold method” was used for preparation of poloxamer gels [22] and modified to accommodate high content of hydrophobic drug where required. Briefly, appropriate amount of TBN was dissolved in sufficient quantity of ethanol followed by addition of the calculated amount of P407. Whenever quantity of ethanol in final formulation was less than the quantity of ethanol required to dissolve TBN, excessive ethanol was used and allowed to evaporate up to required level (semisolid mass/thin film was formed on the walls and bottom of the container depending upon the amount of ethanol). Cold thioglycolic acid aqueous solution (having calculated TGA and water) was added to semisolid mass/thin film slowly and placed for 24 h in refrigerator or freezer whatever required. The final gel formulation had the required concentration of P407, TGA, ethanol and TBN.

### 2.6. Estimation of Drug Loading in Gels

The drug content of gels was increased slowly with increments of 1 mg/g until the precipitates of drug in the gel were visible with naked eye. The gel samples below and above the saturation point were examined under microscope (Olympus microscope, Model CX41RF, Tokyo, Japan). The effect of increase in P407 (20–30% *w*/*w*), ethanol (3–10% *w*/*w*) and modification in method on drug loading was investigated.

### 2.7. Experimental Design

Response surface method based on FCCCD was used to investigate relationship between independent variable on responses which involved full factorial along 6 replicates in the center. Response surface methodology (RSM) helps in identifying the significant variables, best process conditions and to study the interaction between key variables and responses with fewer experiments. The levels of the studied factors were selected so that they were within practical use and their relative difference was adequate to have a measurable effect on the response. A design consisting of 20 runs of experiments was generated using Design-Expert^®^ 6.0.6 software (State Ease Inc., Minneapolis, MN, USA). Independent variables employed were concentration of P407 concentration (X1) with a constraint of 30% *w*/*w*, ethanol (X2) with a constraint of 10% *w*/*w* and thioglycolic acid (X3) with a constraint of 10% *w*/*w*. Constraint was identified based on preliminary experiment and literature. Below 20% *w*/*w*, P407 solution was in liquid form, and above 30% *w*/*w*, it was difficult to dissolve. Similarly, maximum ethanol concentration was kept at 10% *w*/*w* due to its undesirable effect on gel strength. TGA was kept in a concentration limit which is considered safe (below 15% for topical use). The dependent variables were gelation temperature (Y1) and gel strength (Y2).

### 2.8. Determination of Gelation Temperature

The sol-gel transition temperature of the P407 solutions was evaluated by visual tube inversion method as reported previously [23]. The glass tube containing 1 g of the sample was kept in sample cooler (Shimadzu Corporation, Kyoto, Japan). The temperature was gradually increased and the temperature at which no flow of solution occurred after tube inversion was noted as the gelation temperature (*t*_1_). After raising the temperature well above the *t*_1_ and maintaining for 15 min, the temperature was then lowered and the temperature, at which flow of the gel started, was noted (*t*_2_). The mean ± SD of *t*_1_ and *t*_2_ is reported here as the gelation temperature.

### 2.9. Determination of Gel Strength

The strength of poloxamer gel formulations was determined to optimize gel consistency [24]. A glass tube assembly (25 g) which was hollow from inside was placed on surface of gel (1 g) in a glass tube at 32 °C. Gel strength was measured by recording the time (s) taken by the glass tube to penetrate 1 cm into the P407 gel.

### 2.10. Rheology

The gel formulation was subjected to rheological characterization using a Brookfield rheometer (DV3T, Middleboro, MA, USA). Spindle CPA-52Z was employed to determine the viscosity in centipoises at 32 ± 2 °C and the flow pattern was studied by constructing the graph. The graphs were presented as apparent viscosity (η) to the function of shear rate (s-1). The experiment was performed in triplicate and reported as mean ± SD.

### 2.11. Washability/Erosion Profile of P407 Gel

The erosion profile of prepared gel was measured by gravimetric method [25] with slight modification. Briefly, gel formulation (1 g) in small glass tube was placed in a beaker containing water at 32 °C. To the glass tube, 3 mL of phosphate buffer (PB) pH 5.5 maintained at 32 °C was added. At regular interval, the amount of eroded gel was determined by noting reduction in weight of gel after removing the dissolved liquid. One cycle of solvent addition and removal of dissolved material was taken as one washing and number of washings required to remove the percentage of gel from glass tube were noted.

### 2.12. In Vitro Drug Release

Dialysis membrane diffusion model was used to determine in vitro release profile of TBF from optimized P407 gel [15]. Briefly, gel formulation equivalent to 5 mg of TBN was added to cellulose membrane (molecular weight of 12–14 kDa). The membrane was placed in 50 mL of PB (pH 5.5): ethanol (9:1 ratio) in beaker at 32 °C and stirred at 50 rpm using magnetic stirrer. Samples were taken at regular intervals, replenished by the fresh medium to maintain the sink condition. The amount of TBN released was quantified using UV spectrophotometer at 283 nm as reported previously [26] after constructing the standard curve (y = 0.0181x − 0.0015, R^2^: 0.998) using TBN solutions of 1.56 to 50 µg/mL. In vitro release study of drug suspension was performed in a similar manner. All experiments were performed in triplicate (*n* = 3) and results presented as mean ± SD.

### 2.13. Kinetic Modelling of Erosion Profile and In Vitro Drug Release

Gel erosion profile and in vitro drug release data was evaluated to predict release kinetics and mechanism of drug release by applying mathematical models, i.e., zero order, 1st order, Korsmeyer–Peppas, Higuchi and Hixson–Crowell model using DDSolver, a Microsoft Excel Add-In program. Most suitable model was selected based on the goodness of fit test (calculation of R2). *n* value obtained from Korsmeyer–Peppas model was used to find mechanism of drug release.

### 2.14. In Vitro Drug Permeation

Permeation of TBN across human nail clippings was performed using Franz diffusion cell [15]. Nail clipping of known thickness was mounted between donor and receiver compartment of Franz cell. Interface of donor and receiver compartment was occluded with parafilm. The system was kept at 32 ºC. Receiver compartment was filled with PB (pH 5.5): ethanol (9:1) and checked for leakage. Gel formulation (0.1 g equivalent to 1 mg TBN) was placed on the nail surface and covered the donor compartment with parafilm. Sample was taken after every hour and quantified using UV spectrophotometer at 283 nm as mentioned in 2.12 above. Cumulative amount of the drug (µg) permeating per unit surface area of the nail (cm^2^) was plotted against time (h).

## 3. Results

### 3.1. Screening of Penetration Enhancers Based on Hydration Enhancement Factor

Table 1 shows HEF values obtained after treatment with the aqueous solutions of various chemicals screened as nail penetration enhancers. Reducers of disulphide bonds, i.e., thioglycolic acid and mercaptoethanol, were observed as the best nail penetration enhancers with the highest HEF values of 2.73 ± 0.43 and 1.76 ± 0.47, respectively. Second best penetration enhancers were organic acids, i.e., tartaric acid, oxalic acid, glycolic acid and lactic acid, with HEF values of 1.44 ± 0.07, 1.42 ± 0.15, 1.42 ± 0.15 and 1.32 ± 0.10, respectively. Wetting agents/surfactant also showed slight penetration enhancement (HEF values of 1.23 ± 0.08 and 1.20 ± 0.05 for Tween 20 and P407, respectively). Concentration dependent effect of ethanol was observed, i.e., HEF of 5% (*w*/*v*) solution was 1.06 ± 0.12 while that of 20% solution was 1.24 ± 0.22. Another prominent nail penetration enhancer was resorcinol with a HEF value of 1.39 ± 0.16.

### 3.2. Effect of Penetration Enhancer on the Surface Morphology of Human Nail

Figure 1 shows 1500X and 10,000X magnification scanning electron microscope images of the dorsal surface of nails. The untreated nail shows a relatively smooth surface and compact surface with minor ridges (Figure 1a,d). In the hydrated nail (immersed in deionized water for 72 h), the ridges are prominent (Figure 1b,e). The nail clipping hydrated with TGA solution shows disturbed integrity with the creation of some pore-like appearances (Figure 1c,f).

### 3.3. Effect of Independent Variables on Gelation Temperature of P407 Gel

The formulated P407 gel containing TGA and ethanol was translucent and smooth. P407 gel with different formulation (20 runs as suggested by the Design Expert) showed varied gelation temperature ranging from 5.25 to 20.4 °C (Table 2). Analysis of variance (ANOVA) was carried out using Design Expert software to generate the polynomial equation of the responses. p value of less than 0.05 and F value of 164.16 showed that the quadratic model was significant to investigate the effect of dependent variables on gelation temperature. The “Predicted R-Squared” of 0.9488 is in reasonable agreement with the “Adj R-Squared” of 0.9872. This means, 94% variation in response will be explained by this model. Coefficient of variation for the suggested model was 3.58%.

“Adequate Precision” measures the signal to noise ratio. Here, the ratio of 48.49 indicates adequate signal. The “Lack of Fit *F*-value” of 3.95 implied that the Lack of Fit is not significant. Appendix A shows that predicted and actual experimental results have an acceptable agreement which shows less residual and no significant error. Appendix A confirmed normal distribution of residuals in the plot of residuals versus the predicted response due to closeness of points to the straight line.

Values of “Prob > F” less than 0.05 indicate model terms are significant. In this case, A, B, C, C2 are significant model terms. The equation fitted to the data was:Y_1_ = 480.72 - 4.43A + 1.21B − 2.38C + 0.2AB + 0.3AC − 0.3BC + 2.04A^2^ + 0.4B^2^ + 0.96C^2^(2)
where A, B and C is the concentration (% *w*/*w*) of P407, ethanol and TGA, respectively. A positive sign in front of the terms shows synergistic effects while the negative sign indicates the inverse relation between the factors. The polynomial equation for Y1 showed that concentration of polymer has the most significant effect (regression coefficient = −4.43, F value = 911) on gelation temperature followed by thioglycolic acid (regression coefficient = −2.38, F value = 264) and ethanol (regression coefficient = 1.21, F value = 68). A significant (*p* < 0.05) decrease in gelation temperature was observed with increase in quantity of both P407 and TGA (Table 2 and Figure 2). However, elevation of gelation temperature was observed when ethanol concentration was increased (Table 2 and Figure 2).

### 3.4. Effect of Independent Variables on Gel Strength

The gel strength of 20 different runs of P407 formulation was found in the range of 69 to 142 s. *p* value of 0.05 and F value of 60.26 showed that the quadratic model was significant to investigate the effect of dependent variables on gel strength and response optimization. The “Predicted R-Squared” of 0.8672 was in reasonable agreement with the “Adj R-Squared” of 0.9656. This means, 86% variation in response will be explained by this model. Coefficient of variation for suggested model was 3.58%.

“Adequate Precision” measures the signal to noise ratio. A ratio greater than four is desirable. Here, the ratio of 28.19 indicates adequate signal. The “Lack of Fit *F*-value” of 8.92 implied that the Lack of Fit is significant. This occurred due to noise. Appendix A showed that predicted values were close to the actual experimental value which shows less residual and no significant error. Appendix A confirmed normal distribution of residuals in the plot of residuals versus the predicted response due to closeness of points to the straight line.

Values of “Prob > F” less than 0.05 indicate model terms were significant. In this case, A, B, C, BC, A2, C2 were significant model terms.

Following polynomial equation was generated to investigate the effect of independent variables on gel strength.
Y_2_ = 122.9 + 28.13A − 5.47B + 8.17C + 1.0AB + 2.92AC + 0.5BC − 4.3A^2^ − 1.3B^2^ − 6.47C^2^(3)
where A, B and C is concentration (% *w*/*w*) of P407, ethanol and TGA, respectively. The polynomial equation for Y_2_ showed that concentration of polymer (regression coefficient = +28.13, F value = 450) has the most significant effect on gel strength followed by thioglycolic acid (regression coefficient = +8.17, F value = 37.9) and ethanol (regression coefficient = −5.47, F value = 17.02). As depicted in Table 2 and Figure 3, gel strength was increased when quantity of both P407 and TGA was increased. *p* value of less than 0.05 showed that effect of both P407 and TGA was significant. Reduction in gel strength of P407 gel was observed with an increase in concentration of ethanol as depicted in Table 2 and Figure 3.

### 3.5. Response Optimization

Numerical optimization was performed to select formulation with gelation temperature below room temperature and enough gel strength. Constraints applied were to minimize gelation temperature and maximize gel strength. Optimum gel formulation with desirable properties (gelation temperature below room temperature and maximum strength) contains 27.32% P407, 3.18% ethanol and 4.72% TGA. Values of gelation temperature and gel strength predicted by quadratic model were 9.89 °C and 115 s, respectively. The model was validated by formulating optimum formulation and observing its response. The actual gelation temperature and the gel strength of the optimized formulation were 8.68 °C and 122 s, respectively, which were close to the predicted values.

### 3.6. Effect of Preparation Method on Loading of Drug

Figure 4 shows the saturation limit of TBN loading in different formulations. For the gels prepared with cold method of preparation, the maximum loaded concentration of TBN in P407 gels (with no cosolvent and penetration enhancer) was 0.1, 0.2 and 0.4% *w*/*w* for gels containing 20% *w*/*w* (P20), 25% *w*/*w* (P25) and 30% *w*/*w* (P30) of P407, respectively. Similarly, maximum loaded concentration of TBN in gel containing 27% *w*/*w* of P407 having ethanol in concentration of 3% *w*/*w* (PE3), 6% *w*/*w* (PE6) and 10% *w*/*w* (PE10) was 0.3, 0.4 and 0.5% *w*/*w*, respectively. When the modified method was used, the maximum loaded drug in optimized gel was somewhere between 1.2 and 1.5% *w*/*w* (PE3F). No precipitate was observed until a concentration of 1.2 and at 1.5% *w*/*w* of TBN, precipitation of TBN was observed with naked eye. Microscopic images of optimized formulation with clear crystalline precipitates of insoluble TBN in P407 gel (with 1.5% of TBN) is shown in Figure 5.

### 3.7. Rheology

The graph presented in Figure 6 shows apparent viscosity (η) of the gel to the function of applied shear rate (s^−1^). It was observed that apparent viscosity of the gel decreased when applied shear rate was increased, which reflects that optimized gel will behave as non-Newtonian (pseudoplastic) type fluids.

### 3.8. Washability/Gel Erosion

As shown in Figure 7, 76.46 ± 1.6% of the gel was retained after six washings while 23.54 ± 1.6% of P407 gel was eroded. After 12 washings, 52.25 ± 3.48% of the gel remained while 47.75 ± 3.48% of P407 gel was eroded after 12 washes, respectively.

### 3.9. In Vitro Release Profile

In vitro release profile of TBN loaded P407 gel showed that 44.23 ± 1.10 and 67.21 ± 2.16% of the drug was released after 6 and 12 h, respectively (Figure 8). The cumulative release was found to be >90% (94.36 ± 2.26%) in the case of drug suspension in just three hours.

### 3.10. Kinetic Modelling of Erosion Profile and Drug Release

Table 3 shows the kinetic parameters of erosion and drug release profile of optimized gel formulation. It appears that erosion of gel fits to zero order kinetics (R2 = 0.9989). The drug release data was fitted best to the first order kinetics (R2 = 0.9918) which showed that drug release was dependent on concentration of drug remaining in the formulation. Anomalous behavior of drug release was observed as suggested by *n* value of 0.67 obtained from the Korsmeyer–Peppas model which suggests that release of drug from gel may be through both diffusion and erosion of gel.

### 3.11. In Vitro Drug Permeation

The permeation of TBN obtained after placing optimized gel formulation with and without PE on the nail sample between donor and receiver compartment using Franz diffusion cell is shown in Figure 9. Concentration of TBN in receiver fluid was increased with the passage of time. The cumulative amount of TBN permeated from P407 gel with and without PE after 24 h was 27.30 ± 4.18 and 16.69 ± 2.31 µg/cm^2^, respectively.

## 4. Discussion

In the present study, permeation enhancers were screened based on HEF for their ability to enhance nail permeation. Weight of the human nail, when placed in water, increases due to ingress of water into the nail. Chemicals which cause any structural or physicochemical alteration in the nail will affect their ability to absorb water. Disulphide bonds in nail keratin are responsible for the nail barrier property so reducers of disulphide bonds will incorporate changes in nail keratin leading to the creation of pores in the nail which will increase the ability to hydrate the nail [27], as shown by the highest HEF for thioglycolic acid and mercaptoethanol. SEM analysis in our study confirms the formation of pores which create a pathway for permeant to penetrate intermediate and ventral layers of the nail plate. Tween 20 and P407 showed some penetration enhancing properties because they reduce surface tension in water-filled pores of the nail and enhance wetting [19]. Ethanol solution showed a concentration dependent increase in HEF; the higher the concentration of ethanol, the higher the HEF was. Khengar et al., 2007, observed that ethanol resulted in a very small amount of nail swelling which is in accordance with our finding [28].

Poloxamer polymer possesses surfactant properties [29] which is expected to help in nail penetration and solubilization of hydrophobic TBN. However, the solubilizing property of P407 aqueous solution was insufficient to hold the required dose of TBN (1% *w*/*w*). TBN was added to aqueous P407 solution, but this system was unable to solubilize the drug completely even at a concentration of 30% *w*/*w* P407, as shown in Figure 4. Ethanol was used as a cosolvent because of the higher solubility of TBN (up to 45 mg/mL) [30] and P407 polymer in ethanol [31]. However, P407 gel containing 10% *w*/*w* ethanol was also incapable of loading 1% w/w of TBN (Figure 4). As there was insignificant nail penetration enhancement by ethanol, our goal was to keep the quantity of ethanol as minimal as possible due to its undesirable effect on gel strength [31]. Film hydration method was employed as this method gave us the minimum amount of ethanol (3.18%) in the final formulation with enough drug loading (up to 1.2% *w*/*w*), as shown in Figure 4. Guiliano et al., 2020, incorporated the hydrophobic drug rutin to P407 gel prepared by cold method but was unable to load more than 0.1% *w*/*w* rutin, even in the presence of 2% *w*/*w* ethanol [32]. In a similar study to load hydrophobic drug doxorubicin, Xuan et al., 2011, prepared doxorubicin loaded P407 (15% *w*/*w*) and P188 (6% *w*/*w*). HCl (0.1% *w*/*w*) was used as a solubilizer of doxorubicin but it managed to load only 0.6% *w*/*w* of drug [33]. As poloxamer hydrogel has limitations of insufficient gel strength, weak stability and fast dissolution of the gels [34], both additives (TGA and ethanol) and P407 were investigated for their effect on gelation temperature and gel strength to overcome the soft gel properties with adequate loading of hydrophobic TBN. Thioglycolic was used in concentration of 4.72% *w*/*w* as this is claimed to be safe at a concentration of 15% and also sufficient to enhance permeation of TBN [35].

Design of experiments (DOE) is a widely employed organized method to determine the relationship between factors that influence outputs of a process. The DOE approach reduces the number of experiments and detects optimal response within the experimental space. Central composite design (CCD) is the most widely used response surface design. CCD design has advantages of less numbers of tests, high precision and good predictability. The FCCCD is recommended in many practical situations when the specified ranges of levels on the design variables are strict. In other words, the region of interest and the region of operability are the same [36].

The selected model (quadratic) for both responses showed a good fit with the experimental data, which was confirmed by high R^2^ values and *p* value. Furthermore, “Adequate precision” is frequently used to assess signal to noise ratio (predicted response related to its associated error) and this ratio of greater than four is usually desirable. Gelation temperature and gel strength showed high signal to noise ratio, indicating the high adequacy of the selected model.

Appendix A show the predicted versus actual plots for gelation temperature and gel strength, which show closeness of the predicted value to the actual one. Each plot enables evaluation of the capability of the model for prediction. As the predicted values come closer to the actual ones, the points on the scatterplot fall closer to the line. If the points are all very close to the line, the model is expected to offer good predictability. Studentized residuals were distributed along a straight line with a slight deviation confirming low residual and non-significant error. In general, a coefficient of variation (CV) value of less than 10% normally signifies the reproducibility of the generated quadratic model. Relatively lower CV values recorded from the study (4.48% gelation temp, 3.58) confirm the reliability and accuracy of the model. If there is an inefficiency in the model in representing the data, this can be measured from the lack of fit value which was non-significant for gelation temperature but significant for gel strength, which could be due to noise. All these properties suggest that the quadratic model can be used to investigate the effect of P407, TGA and ethanol on gel properties.

The hydrophobic interaction of P407 copolymer chains is responsible for its thermogelation property. These copolymer chains exist in unimer form at concentration below critical micelle concentration (CMC) but start to self-assemble into a spherical micellar structure when its solution concentration exceeds the CMC or the solution temperature is increased above the critical micelle temperature (CMT) [31,37]. The formation of micelle occurs due to dehydration of the hydrophobic polypropylene oxide repeat units and defines the first step of gelation [38,39]. The core of micelles consist of a hydrophobic polypropylene oxide central core with their hydrophilic polyethylene oxide chains facing the external medium. When P407 concentration exceeds critical gel concentration, the micelle structures arrange into a lattice [40].

More the quantity of P407, more will be the physical entanglement which causes an increase in viscosity and lowering of the gelation temperature [38]. Ur-Rehman et al. observed that with an increasing concentration of P407, critical micellization temperature is reduced while intensity of micellization is increased [23]. Increasing poloxamer concentration increases the number of polymer chains per micelle up to a plateau value; above plateau value, increasing P407 concentration increases micelle concentration and reduces the average distance between micelles [40]. This results in strong entanglements of hydrophilic corona polyethylene oxide chains of micelles producing hard gels [41]. Addition of an external hydrophobic group reduces gelation temperature and enhances gel strength [42] which could be a possible explanation for the effect of TGA, as increasing its concentration replaces more polar hydroxyl group of water with the less polar thiol group. Further, replacing water with comparatively high molecular weight thioglycolic acid may also contribute to improved gel strength.

Addition of ethanol hinders the close packing of the block copolymer micelles which cause elevation of CMT and gelation temperature [23]. Further, weaker hydrogen bonding of ethanol as compared to deionized water resulted in gel strength reduction [24]. Although the presence of ethanol reduces the gel strength, this property may be exploited to prepare gel containing a very high concentration of P407, which otherwise is very difficult, time consuming and expensive. Initially, ethanol was used to solubilize TBN and P407 and then cold water was added slowly after the evaporation of ethanol. The finalized gel had a very minute amount of residual ethanol which would help in improving the flow of gel during application on nail.

Desirable behavior of gels for transungual applications is pseudoplastic (non-Newtonian) which corresponds to the decrease in apparent viscosity with an increase in shear rate [10]. This is important for easy application at the intended site. At the same time, the gel should be viscous enough in order to stay at the applied location for sufficient time [43].

P407 gel may erode from the nail surface when it comes in contact with water. This will have a two-way impact on topical gel formulation where i) the aqueous gel may not be retained on the nail for a longer period with exposure to accidental water contact and ii) it can be removed while washing with simple water. Our results demonstrate that two accidental washings may erode only 10% of the gel and more than 50% of the gel was available on the surface even after 12 possible accidental contacts of water. Goo et al., 2021, used hyaluronic acid (3.49% *w*/*w*) to enhance gel strength and reduce erosion of P407 gel (23.91% *w*/*w*). However, they observed 100% of gel erosion after seven washings despite the use of hyaluronic acid [44]. Cleansing of nails to remove nail polish or nail lacquer requires organic solvents which are not friendly to nails and the environment [45]. This study demonstrates that simple rinsing without rubbing may remove plenty of gel and erosion of gel is independent of the amount left on the nail. In the current study, erosion profile data were best fitted to zero order kinetics; however, drug release profile followed first order kinetics. This discrepancy might be due to the presence of ethanol in the release medium responsible for diffusion of drug, which is in accordance with the findings of Wang et al. [46]. The Korsmeyer–Peppas “*n*” value of 0.67 showed non-Fickian anomalous diffusion which suggests both diffusion and erosion were responsible for drug release [47]. It was observed that the presence of TGA enhances the permeation of TBN across the nail which may be due to breakage of nail disulphide bonds responsible for the nail barrier property [48]. Permeated drug concentration (7.6 µg/mL) was above minimum inhibitory concentrations (0.001–0.01 µg/mL) and minimal fungicidal concentrations (0.003–0.006 µg/mL) were against dermatophytes [49]. Gel formulation without TGA also showed permeation of TBN although less than that with TGA containing gel. It can be attributed to the nail uptake enhancing property of ethanol [48]. Rapid permeation of ethanol alters the solubility property of nail resulting in enhanced partitioning of drug into the nail. Optimized formulation with enough drug loading and gel strength.

## 5. Conclusions

Soft gel property and low drug loading of a water insoluble drug hinders the application of P407 gel in transungual delivery of hydrophobic drugs. TGA (up to 5%) can safely be added to P407 gels as a nail penetration enhancer. In addition, by applying a minor modification to the method of preparation, an evaporable cosolvent such as ethanol can be used to load the required dose of hydrophobic drugs which makes P407 gel a promising carrier for transungual delivery of hydrophobic antifungals in onychomycosis.

## Figures and Tables

**Figure 1 polymers-13-03376-f001:**
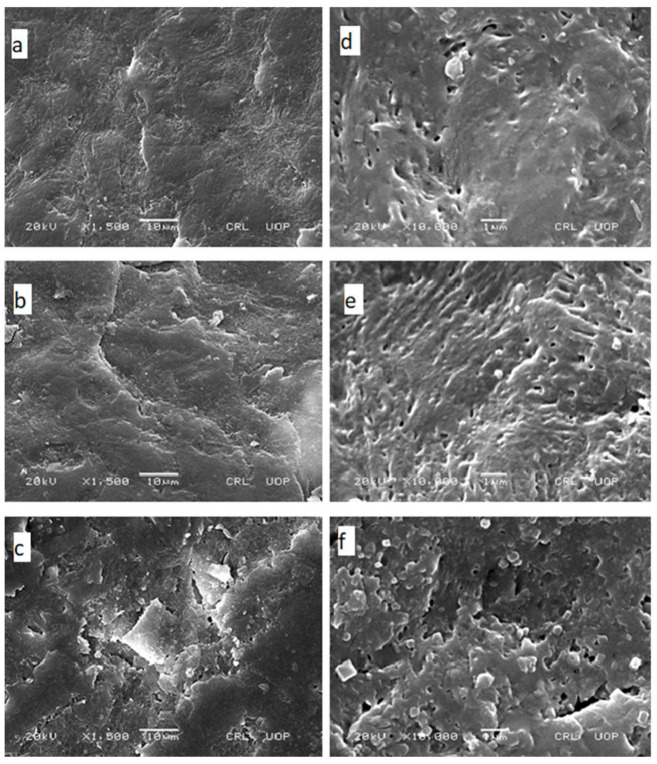
The impact of penetration enhancer on the surface morphology of nail observed under the electron microscope at low (1500×: left-side images) and high (10,000×: right-side images) magnification. Untreated nail (**a**,**d**), nail hydrated in water for 72 h (**b**,**e**) and nail hydrated in 5% TGA solution for 72 h (**c**,**f**).

**Figure 2 polymers-13-03376-f002:**
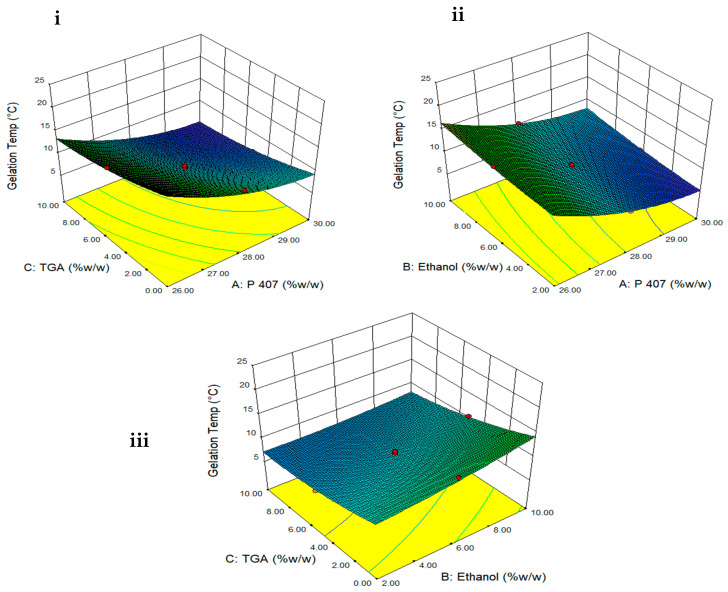
Three-dimensional surface plots showing effect of (**i**) TGA and P407 concentration on gelation temperature when ethanol is constant, (**ii**) ethanol and P407 concentration on gelation temperature when TGA is constant and (**iii**) TGA and ethanol concentration on gelation temperature when P407 is constant.

**Figure 3 polymers-13-03376-f003:**
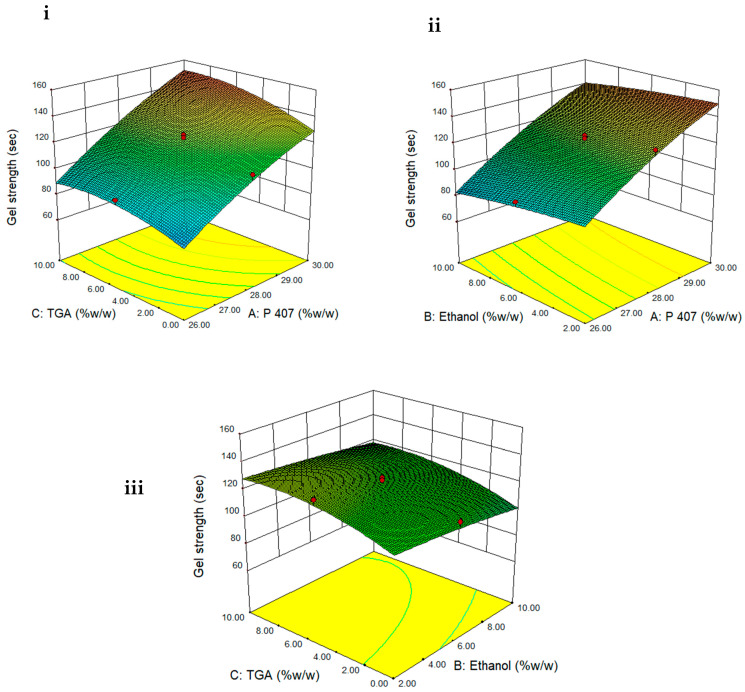
Three-dimensional surface plots showing effect of (**i**) TGA and P407 concentration on gel strength when ethanol is constant, (**ii**) ethanol and P407 concentration on gel strength when TGA is constant and (**iii**) TGA and ethanol concentration on gel strength when P407 is constant.

**Figure 4 polymers-13-03376-f004:**
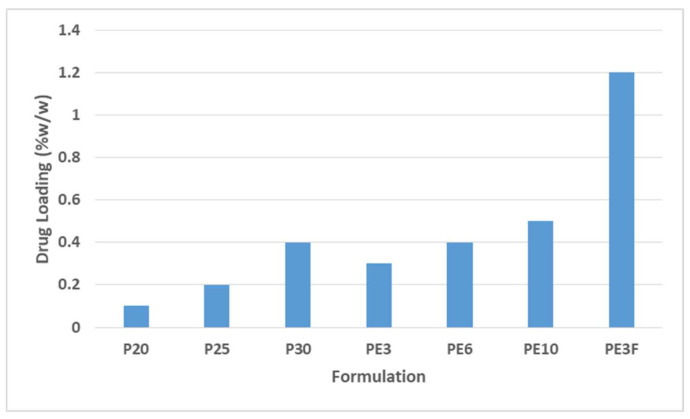
Saturation limit of terbinafine HCL loading in different formulations. P20, P25, P30 (simple P407 aqueous gels with increasing concentration of P407) and PE3, PE6, PE10 (gels containing 27% P407 with increasing concentration of ethanol) were prepared with simple cold method whereas PE3F is optimized formulation (27.32% of P407, 3.18% of ethanol and 4.76% of glycolic acid) prepared with film hydration method.

**Figure 5 polymers-13-03376-f005:**
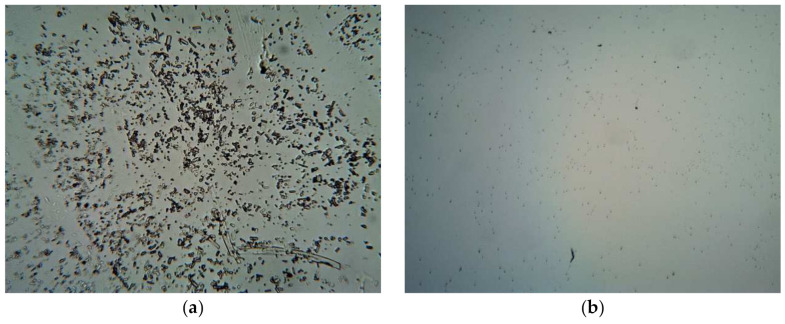
Images of optimized formulation under light microscope at 10X: (**a**) gel containing 1.5% w/w of TBN and (**b**) gel containing 1.2% *w*/*w* of TBN.

**Figure 6 polymers-13-03376-f006:**
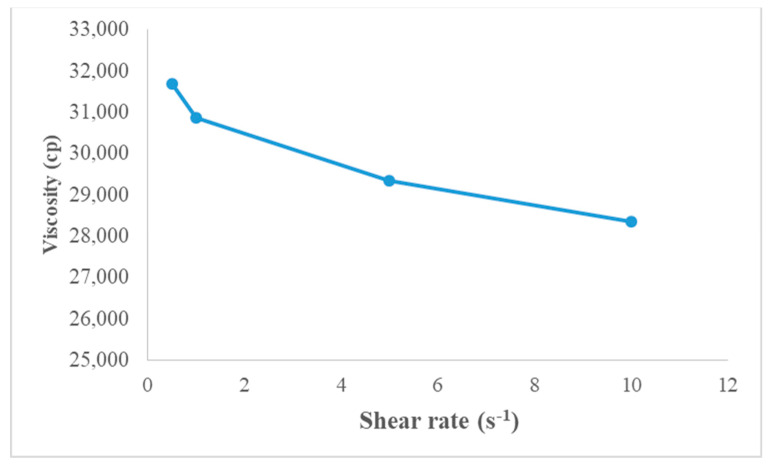
Apparent viscosity (η) vs. shear rate of P407 gel.

**Figure 7 polymers-13-03376-f007:**
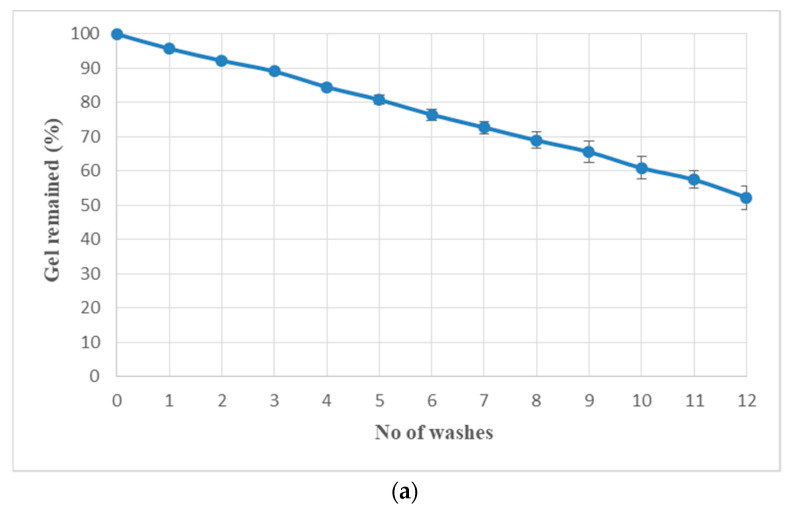
(**a**) Washability and (**b**) gel erosion pattern of P407 gel formulation in phosphate buffer.

**Figure 8 polymers-13-03376-f008:**
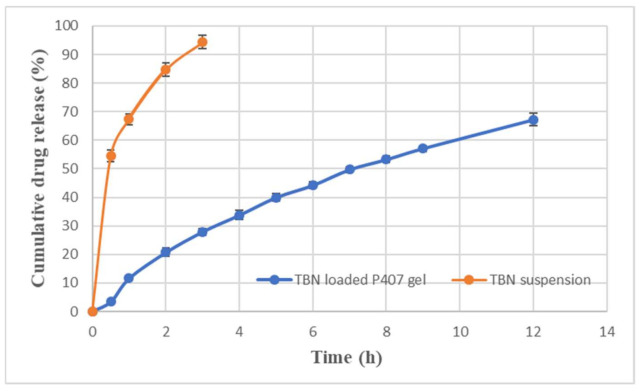
Comparison of in vitro release of terbinafine from optimized P407 gel formulation and TBN suspension.

**Figure 9 polymers-13-03376-f009:**
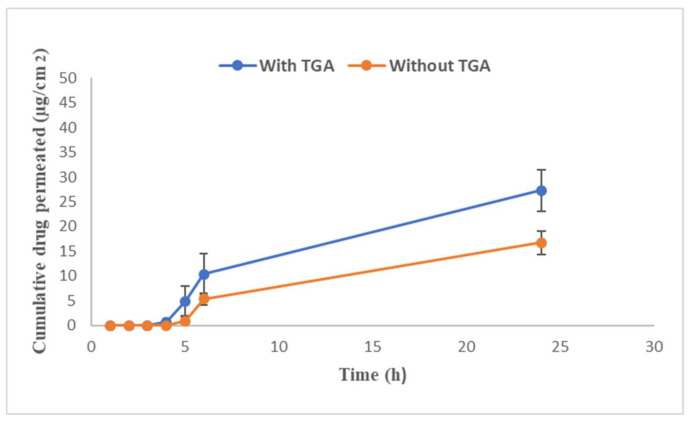
In vitro drug permeation across human nail.

**Table 1 polymers-13-03376-t001:** Comparison of HEF values of screened chemicals as potential nail penetration enhancers.

S.No.	Penetration Enhancer	Concentration in the Nail Treatment Solution % *w*/*v*	Hydration Enhancement Factor (HEF) Mean ± SD
1.	Thiourea	5	1.19 ± 0.06
2.	Sodium lauryl sulphate	5	0.91 ± 0.13
3.	Tween 20	5	1.23 ± 0.08
4.	Tween 80	5	1.14 ± 0.21
5.	DMSO	5	1.18 ± 0.46
6.	Oxalic acid	5	1.42 ± 0.15
7.	Urea	5	1.04 ± 0.19
8.	Thioglycolic acid	5	2.73 ± 0.43
9.	Glycolic acid	5	1.42 ± 0.15
10.	Mercaptoethanol	5	1.76 ± 0.47
11.	Resorcinol	5	1.39 ± 0.16
12.	Alpha cyclodextrin	5	1.22 ± 0.18
13.	Hydroxy Propyl-β-Cyclodextrin	5	0.87 ± 0.08
14.	Chitosan	1	1.08 ± 0.28
15.	Chitosan	0.5	0.86 ± 0.05
16.	Thiolated chitosan	0.5	1.23 ± 0.17
17.	β-Cyclodextrin	5	1.09 ± 0.13
18.	Tartaric acid	5	1.44 ± 0.07
19.	Methionine	5	0.98 ± 0.12
20.	Lactic acid	5	1.32 ± 0.10
21.	Ethanol	5	1.06 ± 0.12
22.	Ethanol	20	1.24 ± 0.22
23.	Poloxamer 407	5	1.20 ± 0.06

**Table 2 polymers-13-03376-t002:** Different runs of face-centered central composite design and corresponding responses.

Code	P407 (% *w*/*w*)	Ethanol (% *w*/*w*)	TGA (% *w*/*w*)	Gelation Temp (°C)	Gel Strength (s)
F1	26	2	10	13.46 ± 0.28	92.66 ± 5.13
F2	28	6	5	8.43 ± 0.28	122.33 ± 8.50
F3	30	10	0	11.85 ± 0.25	124.66 ± 9.50
F4	26	6	5	15.46 ± 0.18	93.66 ± 6.02
F5	30	2	10	5.25 ± 0.16	156 ± 7.21
F6	28	6	10	6.53 ± 0.09	119 ± 8.54
F7	28	10	5	10.4 ± 0.32	111 ± 7.54
F8	26	10	0	20.43 ± 0.23	69 ± 7.54
F9	30	10	10	6.33 ± 0.09	149 ± 9.16
F10	30	6	5	5.93 ±0.23	142.33 ± 6.65
F11	30	2	0	7.38± 0.35	131 ± 7.54
F12	28	6	5	8.53 ± 0.14	122 ± 4.35
F13	28	6	5	8.66 ± 0.16	121.33 ± 9.29
F14	26	2	0	17.35 ± 0.25	82 ± 5.56
F15	28	6	5	8.61 ± 0.32	124.3 ± 7.02
F16	28	6	5	9.2 ± 0.23	126.33 ± 7.57
F17	28	6	0	12.7 ± 0.21	112.66 ± 5.50
F18	28	2	5	7.71 ± 0.21	131 ± 8.88
F19	28	6	5	8.38 ± 0.22	124.33 ± 4.93
F20	26	10	10	14.3 ± 0.32	84.33 ± 5.50

**Table 3 polymers-13-03376-t003:** Kinetic modelling of erosion profile and in vitro drug release.

Model		For Gel Erosion	For Drug Release
Zero order	K	3.89	6.60
	R^2^	0.9989	0.8894
First order	k1	0.048	0.098
	R^2^	0.9814	0.9918
Higuchi	kH	11.01	18.14
	R^2^	0.8142	0.9475
Korsmeyer–Peppas	kKP	3.72	12.99
	R^2^	0.9990	0.9902
	*n*	1.02	0.67
Hixson–Crowell	kHC	0.015	0.02
	R^2^	0.9898	0.9769

## Data Availability

Not applicable.

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
