# Peer review of "Poloxamer 407 Based Gel Formulations for Transungual Delivery of Hydrophobic Drugs: Selection and Optimization of Potential Additives"

_polymers, 2021, doi:10.3390/polym13193376_

Round 1

Reviewer 1 Report

  1. The manuscript heavily focuses on formulation optimization, however, in order to demonstrate efficacy, the authors should depict some more information/studies on how this optimized formulation performs with other comparators.
  2. The authors must provide a reference to the UV method they adopted. They should also state any modifications in the method, if any.
  3. There is no discussion for the Experimental design analysis. The authors just repeated what was mentioned in the equations/tables. Please add a reasonable explanation for the results.
  4. Please add lack of fit and residual /diagnostic analysis with regard to FCCCD Experimental design analysis
  5. Manuscript stated that the concentration of polymer has most significant effect on gelation temperature followed by thioglycolic acid and ethanol. Please add a reasonable explanation for the results.
  6. As stated in aim to improve transungual permeation; then transungual permeation could have been taken as one of the response/dependent variable in the study.
  7. The introduction states that " topical P407 gel formulation was developed with the aim to enhance the loading of hydrophobic drugs and to improve transungual permeation without compromising on thermo-gelation and washability of these gels.". However, this statement is not addressed by the end of the manuscript as there is no work evaluating loading of hydrophobic drugs in gel and also study where comparision or improvement in transungual permeation.
  8. Manuscript stated that gel strength was increased when quantity of both P407 247 and TGA was increased. Please add a reasonable explanation for the results.
  9. Check the lines 245 and 225. Both discussing the same and seems repetitions.
  10. Manuscript stated that the graph presented in Fig. 4 shows apparent viscosity (η) of the gel to the function 268 of applied shear rate (s-1); but the manuscript shows D surface plots.
  11. Please explain the basis for selection of level for selected independent variables.
  12. Any reason for selection of FCCCD Experimental design over other available Response Surface Designs.
  13. The authors need to build more discussion on the optimized formulation.
  14. Safety concerns regarding usage of thioglycolic acid should be supported by literature.
  15. How did the authors identify the constraints on their variables? Please consider incorporating in the manuscript.
  16. Some useful references considering this manuscript; https://doi.org/10.1016/j.ijpharm.2018.11.016, https://doi.org/10.1016/j.ijpharm.2019.06.008, https://doi.org/10.1208/s12249-021-01950-x,

Author Response

Reviewer #1

Thank you for your valuable suggestions and comments. Your review and comments on our manuscript improved the quality of our manuscript. Following is the numberwise reply to your valuable comments.

  1. The manuscript heavily focuses on formulation optimization, however, in order to demonstrate efficacy, the authors should depict some more information/studies on how this optimized formulation performs with other comparators.

Response: According to your valuable suggestion, we have compared our optimized formulation with the previous studies in the discussion section. Following details were added.

“Film hydration method was employed as this method gave us minimum amount of ethanol (3.18%) in final formulation with enough drug loading (upto 1.2%w/w) as shown in Fig. 2. Guiliano et al., 2020 incorporated hydrophobic drug rutin to P407 gel prepared by cold method but was unable to load more 0.1% w/w rutin even in the presence of 2%w/w ethanol 32 . In a similar study to load hydrophobic drug doxorubicin, Xuan et al., 2011 prepared doxorubicin loaded of P407 (15%w/w) and P188 (6%w/w). HCl (0.1%w/w) was used as solubilizer of doxorubicin but they managed to load only 0.6% w/w of drug ”.

“Our results demonstrates that two accidental washing may erode only 10% of the gel and more than 50% of the gel was available on the surface even after 12 possible accidental contacts of water. Goo et al., 2021 used hyaluronic acid (3.49%w/w) to enhance gel strength and reduce erosion of P407 gel (23.91% w/w). However they observed 100% of gel  erosion after 7 washings despite the use of hyaluronic acid”.

  1. The authors must provide a reference to the UV method they adopted. They should also state any modifications in the method, if any.

Response: Following your instructions, the reference for UV method for quantification of  terbinafine has been added in the manuscript (Reference no 26 added to section 2.12).

  1. There is no discussion for the Experimental design analysis. The authors just repeated what was mentioned in the equations/tables. Please add a reasonable explanation for the results.

Response: Following your kind suggestion, the result of the experimental design has been discussed and added to the discussion part as under.

“Design of experiments (DOE) is a widely employed organized method to deter-mine the relationship between factors that influences outputs of a process. DOE approach reduces number of experiments and detect optimal response within the ex-perimental space. Central composite design (CCD) is the most widely used response surface design. CCD design has the advantages of less number of tests, high precision, and good predictability. The FCCCD is recommended in many practical situations when the specified ranges of levels on the design variables are strict. In other words, the region of interest and the  region of operability are the same”.

  1. Please add lack of fit and residual /diagnostic analysis with regard to FCCCD Experimental design analysis.

Response: Thanks for your valuable recommendation. Lack of fit values has been added section 3.3 and 3.4 in result section. Further residual graphs has been added to depict residual error and actual vs predicted values (fig 1,2,3,4 in supplementary data).

  1. Manuscript stated that the concentration of polymer has most significant effect on gelation temperature followed by thioglycolic acid and ethanol. Please add a reasonable explanation for the results.

Response: Following your suggestion, result has been explained on the basis of regression coefficient and F value to clear the magnitude of the effect of all independent variables.Following details were added to section 3.3 in result section.

The polynomial equation for Y1 showed that concentration of polymer has most significant effect (regression coefficient = -4.43, F value= 911) on gelation temperature followed by thioglycolic acid (regression coefficient = -2.38, F value= 264) and ethanol (regression coefficient = 1.21), F value= 68) .

  1. As stated in aim to improve transungual permeation; then transungual permeation could have been taken as one of the response/dependent variable in the study.

Response: Thanks for the comment. Your suggestion is very valuable but keeping the permeation as response variable was difficult because of arrangement of nail samples for permeation studies(20 runs in triplicate). Further 3Rs of Helsinki accord  state to design experiment to replace or minimize the use of animals in study. So we performed the permeation study for only optimized formulation and there it was compared with gel without TGA.

  1. The introduction states that " topical P407 gel formulation was developed with the aim to enhance the loading of hydrophobic drugs and to improve transungual permeation without compromising on thermo-gelation and washability of these gels.". However, this statement is not addressed by the end of the manuscript as there is no work evaluating loading of hydrophobic drugs in gel and also study where comparison or improvement in transungual permeation.

Response: Thank you for your valueable suggestion. We have added experimental study to demonstrate higher loading of terbinafine by our film hydration method as compare to cold method (section 2.6 and 3.6). Further graph (Fig 4) has been added to compare saturation limit of P407 gel prepared by cold method and film hydration method. The result has also been discussed in the discussion section.

Further, permeation of thioglycolic acid containing gel was higher as compared to gel without thioglycolic acid which showed improvement in permeation.

  1. Manuscript stated that gel strength was increased when quantity of both P407 247 and TGA was increased. Please add a reasonable explanation for the results.

Response: These results has already been explained in the discussion section.

9 . Check the lines 245 and 225. Both discussing the same and seems repetitions.

Response: Thanks for your valueable correction. Line 245 was related to gel strength not gelation temperature. It has been corrected in the revised manuscript in section 3.4.

  1. Manuscript stated that the graph presented in Fig. 4 shows apparent viscosity (η) of the gel to the function 268 of applied shear rate (s-1); but the manuscript shows D surface plots.

Response: Thanks for your valueable correction. Fig 6 is related to viscosity vs shear rate,not Fig 4. It has been corrected in the manuscript in section 3.7.

  1. Please explain the basis for selection of level for selected independent variables.

Response: Following your suggestion, selection of level for independent variables has been explained in the manuscript in section 2.7.

  1. Any reason for selection of FCCCD Experimental design over other available Response Surface Designs.

Response: Thanks for your valuable comment. FCCCD was selected because it has an advantage over the other methods as it is formed by two level factorials by the addition of just enough points to estimate curvature and interaction effects. Further we can exted the number of trials in FCCCD.

  1. The authors need to build more discussion on the optimized formulation.

Response: Thank you for your comment. Follwing your suggestion, optimized formulation has been further discussed and compared with other studies in the discussion section.

  1. Safety concerns regarding usage of thioglycolic acid should be supported by literature.

Response: Thank you for your valauable comment. Thioglycolic acid was used in safe concentration (below 15%) and it was cited in discussion part as under (reference 35).

“Thioglycolic was used in concentration of  4.72%w/w as these are claimed to be safe at concentration of 15% and also sufficient to enhance permeation of TBN”.

  1. How did the authors identify the constraints on their variables? Please consider incorporating in the manuscript.

Response: Thanks for your valuable suggestion. Constrained applied to each independent variable was added in the manuscript with reasons. Following details were added to section 2.7.

“Independent  variables employed were concentration of P407 concentration (X1) with a constraint of 30%w/w, ethanol (X2) with a constraint of 10% w/w, and thiogly-colic acid (X3) with a constraint of 10% w/w. Constraint were identified bases on pre-liminary experiment and literature. Below 20%w/w, P407 solution was in liquid form and above 30%w/w, it was difficult to dissolve. Similarly maximum ethanol concen-tration was kept at 10% w/w due to its undesirable effect on gel strength. TGA was kept in a concentration limit which is considered safe (below 15% for topical use). The dependent variables were gelation temperature (Y1) and gel strength (Y2)”. 

  1. Some useful references considering this manuscript; https://doi.org/10.1016/j.ijpharm.2018.11.016, https://doi.org/10.1016/j.ijpharm.2019.06.008, https://doi.org/10.1208/s12249-021-01950-x,

Response: Thanks for your valuable recommendation.The suggested useful articles have been cited in the manuscript (Reference no 10 and 39)

Reviewer 2 Report

In this manuscript, the authors report a drug delivery system based on well-known poloxamer 407 (P407) gels for the treatment of nail diseases. Here, the gel strength and drug loading capacity of gel are improved using an optimized additive, thioglycolic acid. As a result, the synthesized gel may be useful for transungual drug delivery. The manuscript is well organized/written, and however, the novelty of this work is less. Thus, the authors should address the following comments before publishing the manuscript.

  1. Page 1, line 24; What is the full composition of gel? Which type (weight or volume) of percentage?
  2. Page 1, line 27; The full name of TBN should be included.
  3. In order to demonstrate the novelty, the authors should make a table containing the composition, gel strength, biocompatibility, and drug loading aptitude of gels for comparison of previously reported studies with this work.
  4. The authors should study the in vitro antifungal activity of delivered drugs to improve the novelty.      
  5. The authors should cite the following two recent review articles related to drug delivery.

Medicines 2019, 6, 7; doi:10.3390/medicines6010007

ACS Appl. Polym. Mater. 2021, https://doi.org/10.1021/acsapm.1c00785

Author Response

Thank you for your valuable suggestions and comments. Your review and comments on our manuscript improved the quality of our manuscript. Following is the numberwise reply to your valuable comments.

  1. Page 1, line 24; What is the full composition of gel? Which type (weight or volume) of percentage?

Response: Thanks for your valuable suggestion. The concetration of ingredients of gel was presented in percent w/w and the w/w unit has been added to clear the composition in the abstract of MS. 

  1. Page 1, line 27; The full name of TBN should be included.

Response: Thanks for your valuable recommendation. Terbinafine word was first time used in the abstract  where TBN abbreviation was added. We have used abbreviated form in the rest of the document. Correction has been made.

  • In order to demonstrate the novelty, the authors should make a table containing the composition, gel strength, biocompatibility, and drug loading aptitude of gels for comparison of previously reported studies with this work.

Response:  Thank you for the suggestion. Comparison of drug loading and gel erosion with previous studies have been incorporated in the discussion section of revised MS. In our opinion comparison of biocompatibility will be out of the scope of this MS as it was neither aimed nor performed.

  • The authors should study the in vitro antifungal activity of delivered drugs to improve the novelty.

Response: Thanks for your comment. It is valuable suggestion, however we are working on another MS where antifungal activity of known antifungals and plant extract across the nail is being compared. As TBN is well established antifungal and quantity of terbinafine permeated across the nail plate after application of gel was found to be above the minimum inhibitory concetration against dermatophytes so we think that that permeated drug will prevent the groth of dermatophytes. Furthermore the focus of this MS is P407 based gel formulations so the experimentation of In vitro antifungal activity will be little overdoing in our humble opinion. 

5  The authors should cite the following two recent review articles related to drug delivery.

    Medicines 2019, 6, 7; doi:10.3390/medicines6010007

     ACS Appl. Polym. Mater. 2021, https://doi.org/10.1021/acsapm.1c00785

Response: Thanks for your valuable recommendation. The suggested articles have been cited in the manuscript (Reference no 11 and 37)